# Impact of Environmental Economic Transformation Based on Sustainable Development on Financial Eco-Efficiency

**Limei Yin and Jia Liu \***

School of Economics and Management, Harbin University, Harbin 150086, China
* Correspondence: liujia@hrbu.edu.cn

**Abstract:** The production and life of human beings are inseparable from the natural environment, and the current economic transformation is based on the sustainable development of the environment. However, the current environmental economic transformation lacks a corresponding evaluation model, so this paper aimed to explore the path of environmental economic transformation and analyze the impact of environmental economic transformation on financial eco-efficiency. Aiming at the transformation of environmental economy, this paper analyzed the relationship between the environmental quality and the transformation path and made a detailed analysis of the dynamic and static transformation process. After understanding the path of environmental economic transformation, this paper established a model to analyze the impact of eco-efficiency. In terms of indicators, this paper selected four indicators of environmental economic transformation: return on assets, gross margin of sales, period expense rate, and total asset turnover. Through data analysis, this paper discussed the impact of these four indicators on financial eco-efficiency. The experimental results show that the comprehensive coefficient of environmental and economic transformation indicators is 1.325 ($p < 0.001$). This shows that the environmental economic transformation has a significant positive correlation with the financial eco-efficiency, that is to say, a good environmental economic transformation can increase the financial eco-efficiency index.

**Keywords:** sustainable development; environmental economic transformation; financial eco-efficiency; impact pathway analysis





## 1. Introduction

It has been nearly 40 years since the reform and opening up, and China's economy is also in the transition stage from high-speed to medium high-speed and high-quality. In this context, the current environmental economic transformation lacks the corresponding evaluation model, so this paper aimed to explore the path of environmental economic transformation and analyze the impact of environmental economic transformation on financial ecological efficiency. From the perspective of economic stage transformation, this paper combed the existing theoretical research and literature achievements with the main line of "transformation growth and environmental financial ecological efficiency green consumption transformation". It probes into the relationship between natural environment and economic development in the process of transformation.

The ecological efficiency of various regions in China shows a positive spatial correlation feature, and there is an obvious spatial aggregation feature. "Transformation" is one of the important branches of development economics research. It broadens the research perspective of development economics and finds an ingenious solution to the bottleneck of economic development. It can be said that the topic of "transformation" has been the research focus of the entire economics field in the past half century. Vicol M has studied the dynamics of economic and social change in the little-known highlands of Chin State, Myanmar, in the context of business policy in Southeast Asia [1]. Yongwen N conducted a study on China's 14th Five-Year Economic Plan and discussed the role of China's economic

transformation in promoting the dual circulation in the world [2]. Mavlanov I discussed Uzbekistan's economic and diplomatic transformation in view of the current diplomatic situation in Uzbekistan [3]. Lorenzen M studied the concept and application of forest economic transformation with the forest of Mixteca Alta UNESCO Global Geopark in Oaxaca, Mexico [4]. Against the background that China's tourism industry has made great progress, Lin MJ implemented effective measures to promote economic transformation in order to improve the inherent economic development differences in the dual structure of urban and rural areas [5]. Their research on economic transformation is very comprehensive, not only analyzing the significance of economic transformation, but also analyzing the principles and principles of economic transformation. However, they do not fully consider the environmental issues in economic transformation, and it is difficult to eliminate the impact of environmental factors.

Based on the theoretical basis of eco-efficiency, many companies in the world have performed the practice of eco-efficiency. Zhou C explored the influencing factors of eco-efficiency using a panel data model with fixed effects based on a panel dataset from 2005 to 2014 [6]. In order to simultaneously improve the productivity and eco-efficiency of typical winter wheat-summer maize rotations in the North China Plain, Yue X optimized the interaction of genotype (G) × environment (E) × management (M) and presented best agronomic management practices and cultivars for four representative sites with an agricultural production system simulator model and detailed field trial data [7]. Xing Z combined the economic input-output life cycle assessment and data envelopment analysis to assess the environmental impact and eco-efficiency of 26 economic sectors in China [8]. In view of the current low utilization rate of waste edible oil, Hartini S analyzed the eco-efficiency index of waste edible oil utilization and analyzed the environmental impact of recycling waste edible oil [9]. In order to improve the agricultural eco-efficiency of Henan Province and promote the sustainable development of agriculture, Li B used the super-efficiency SBM model to scientifically calculate and analyze the agricultural eco-efficiency of Henan Province based on the footprint theory [10]. However, since China's theoretical research on eco-efficiency began in the early 21st century, it has only been more than ten years since then, so compared with foreign enterprises, Chinese enterprises have less practice of eco-efficiency, and there is no comprehensive eco-efficiency evaluation model [11,12].

The main contributions and innovations of this paper are as follows: First, this paper built an overall analysis logic and path of economic transformation and environmental quality, and then specifically explained the impact of the transformation process on the environment from the two lines of production transformation and consumption transformation and built a relatively complete research system. Second, starting from the essential causes of environmental pollution, this paper established a capital transformation model that includes the degradation properties of natural capital, and on this basis, discussed the role of natural capital in economic transformation and its impact on economic transformation, and discussed the impact of economic transformation on financial eco-efficiency.

## 2. Environmental Economic Transformation

### 2.1. Evaluation of Eco-Efficiency

Eco-efficiency evaluation is the process of using appropriate indicators to convert the performance of enterprises, such as the degree of energy conservation and emission reduction, economic profit, etc., into simple and understandable information, and it is a systematic procedure for measuring and evaluating the eco-efficiency of enterprises. The goal of eco-efficiency evaluation is to provide effective suggestions for enterprise management, environmental protection and future development. The steps of evaluating eco-efficiency generally include selecting evaluation indicators, establishing an evaluation system, using theoretical models for empirical analysis, drawing conclusions and suggestions, and making improvements based on the current management deficiencies of the enterprise, as shown in Figure 1. Eco-efficiency assessment is an internal management

process and tool. Its ultimate purpose is to continuously provide decision makers with real and reliable information on resource utilization efficiency, environmental protection degree, performance growth rate, etc. Eco-efficiency evaluation is a process that needs to continuously collect information and evaluate the collected information, and the evaluation object is its management system and operating system.

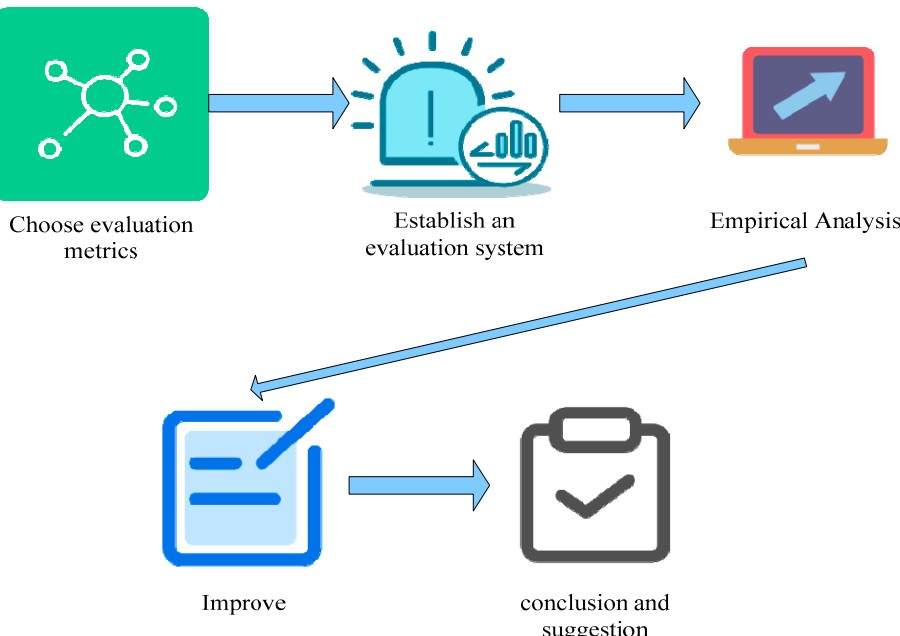

**Figure 1.** Flow chart of eco-efficiency evaluation.

There are generally two reasons why companies conduct eco-efficiency evaluations. First, due to the pressure of the external environment, the evaluation of eco-efficiency has to be conducted, such as pressure from competitors, attention from the public and news media, management from government environmental protection departments, and even mandatory requirements of relevant laws and regulations. Second, the needs of enterprise internal management make the evaluation of eco-efficiency an important part of enterprise strategic management. By evaluating the eco-efficiency, enterprises can discover the existing problems of the enterprise in time, actively adjust the industrial structure, and adhere to the path of sustainable development under the premise of realizing energy saving, emission reduction and win-win economic benefits, and enhance the comprehensive competitiveness of enterprises [13].

### 2.2. Green Transition of Economy

The modern sustainable development theory is a new development concept in the 1980s, which takes into account the needs of the survival and development of the contemporary people and the development needs of future generations and conforms to the requirements of the times and the needs of social and economic development [14,15]. The traditional development model one-sidedly pursues the speed and volume of economic development while ignoring the social and environmental benefits of development, leading to social problems such as inequality of wealth and environmental pollution worldwide [16,17]. The sustainable development theory overcomes the defects of the traditional development model. The green transformation of the economy is based on the theory of sustainable development. The specific green transformation includes green transformation in agriculture, industry, and society.

The process of urbanization has written a difficult course of the continuous development of human society amid contradictions and struggles. As far as cities are concerned, the ubiquitous existence of contradictions is the basis for the steady progress of urbanization,

which is reflected in the continuous competition between the forces of the contradictions related to the urbanization process. Especially after the qualitative change is achieved, the city must make corresponding adjustments—through urban transformation, change the original development model to adapt to the emergence of new contradictions, so as to enter the next development cycle [18]. It can be seen that identifying the stage of urbanization is very necessary to predict whether the contradictory subject can have essential changes and the possible impact after the change. The current urbanization process in China is shown in Figure 2.

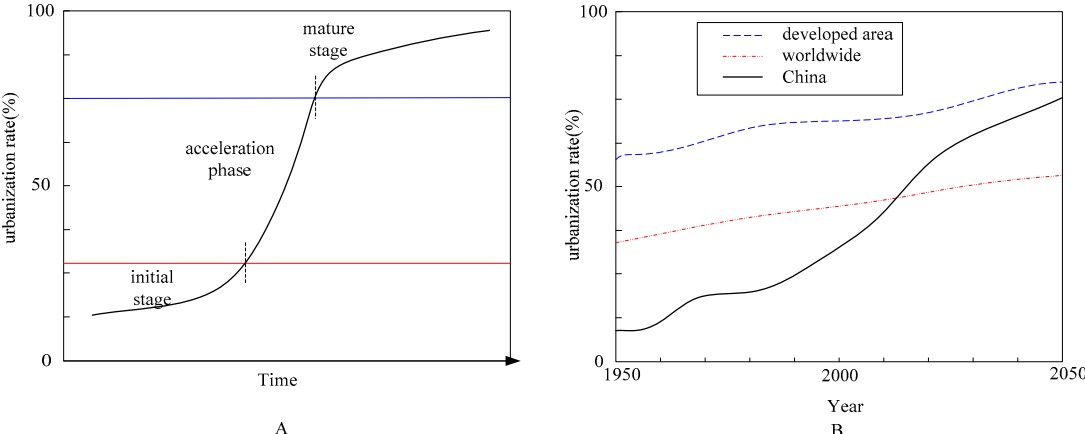

**Figure 2.** China's urbanization process. (**A**). Urbanization rate curve and its corresponding development stages; (**B**). Comparison of urbanization levels and trend forecast.

Figure 2A shows the urbanization rate curve and its corresponding development stages. It can be seen that China's urbanization process is similar to an S-shaped curve. Figure 2B shows the urbanization level comparison and trend forecast. It can be found that China's urbanization level would gradually reduce the distance from developed countries.

### 2.3. Environmental Quality and Economic Transformation Growth Path

2.3.1. Theories and Paths of the Relationship between Environmental Quality and the Transformation of the Three Major Capital Accumulations

Among the three major capitals, natural capital and physical capital are generally considered to have diminishing marginal returns and produce pollution in the production process, while human capital is not only the source of continuous economic growth, but also a symbol of advanced development stage, and it is also a kind of clean "energy" [19]. Diminishing marginal utility refers to when a person continuously consumes a certain kind of goods within a certain period of time, while the consumption quantity of other goods remains unchanged. The total utility would increase with the increase in the quantity of the goods consumed. However, marginal utility of the goods (that is, increase in the utility brought by each unit of the goods consumed) tends to decrease. Therefore, one could first assume that there are two polluting products $X$, $Y$ and one cleaning product $Z$ in a closed economy. This paper used correlation analysis for this research.

The production functions of polluting products $X$ and $Y$ are:

$$X = AN^{\alpha}(0 < \alpha < 1) \tag{1}$$

$$Y = BK^{\beta}(0 < \beta < 1) \tag{2}$$

Among them, $A$ and $B$ are the technical coefficients of producing pollutants, $N$ represents natural capital, $K$ represents physical capital, and $\alpha$, $\beta$ represent the diminishing marginal product of natural capital and physical capital. The production function of the cleaning product $Z$ is:

$$Z = CH \tag{3}$$

There are two stages in the transformation process: The first stage is the transition from natural capital accumulation to material capital accumulation, and the second stage is the process of material capital accumulation to human capital accumulation. The specific transformation process is as follows:

Stage 1: The total output in the economy is

$$G = X + P_1 Y \tag{4}$$

Supposing the capital stock is

$$\overline{K} = N + K \tag{5}$$

$$\alpha A N^{\alpha-1} = \beta B K^{\beta-1} P_1 \tag{6}$$

Then it can be concluded that the critical capital level in the economic equilibrium of Equation (6) is:

$$K_1 = \left( \frac{\alpha A}{\beta B P_1} \right)^{\frac{1}{\alpha-\beta}} \tag{7}$$

The second stage: Similar to the first stage, the total economic volume becomes:

$$G = X + P_1 Y + P_2 Z \tag{8}$$

Solving Equations (1) and (2) to get:

$$P_1 B \beta (\overline{K} - K_1) = P_2 C \tag{9}$$

The capital threshold $K_2$ for the transition from physical capital accumulation to human capital accumulation is obtained as:

$$K_2 = \left( \frac{\beta B P_1}{C P_2} \right)^{\frac{1}{1-\beta}} + K_1 \tag{10}$$

$C$ is constant. When capital $\overline{K} > K_2$ is accumulated, the remaining capital $\overline{K} - K_2$ is used to produce cleaning product $Z$.

### 2.3.2. Dynamic Analysis of the Path of Capital Accumulation in Economic Transformation

In the early stages of the economy, the stock and accumulation of physical capital is extremely scarce. According to the actual development process of European and American countries, the large amount of material capital appeared after the Industrial Revolution, and the amount of material capital was scarce before the Industrial Revolution, which is reflected in the "critical minimum effort" theory and the "big push theory" [20,21]. From the point of view of natural evolution, it also shows that natural capital accumulation is dominated before a large amount of capital accumulation, so in the natural capital accumulation stage, the total level of capital in period t can be calculated by Equation (11):

$$\overline{K}_t = s G_{It-1} = s A N_{t-1}^{\alpha} = s A \overline{K}_{t-1}^{\alpha} \tag{11}$$

If this stage reaches a steady state, the solution is:

$$K^* = (sA)^{\frac{1}{1-\alpha}} \tag{12}$$

From this, the economic transformation can be divided into two stages, as shown in Figure 3:

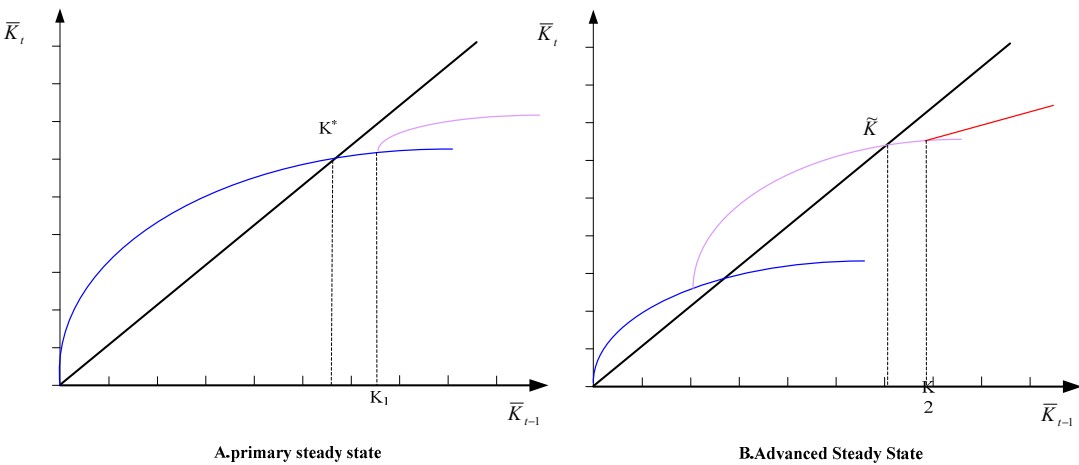

**Figure 3.** Primary Steady State vs Advanced Steady State.

(1) If $K_1 < K^*$, the closed economy has already started the stage of material capital accumulation before reaching a steady state, avoiding falling into the trap of a primary steady state, and successfully crossing the first transitional growth. Although the transformation is successful, products $X$ and $Y$ are still produced at this time, and the environmental quality may deteriorate due to the reduction of the ecological degradation ability of natural capital.

(2) If $K_1 > K^*$, then the economy has entered a primary steady state in the process of natural capital accumulation before the transition. Since natural capital is producing pollutant X and emitting pollutants, it would also cause environmental problems due to the reduction of natural capital's ability to degrade pollutants, making it difficult to maintain the current steady state, and even leading to the collapse of the entire social operation, such as shown in Figure 3A.

If the economy successfully completes the first stage of transformation and enters the stage of material capital accumulation, that is, under the conditions of (1), the same is true for the first stage. Assuming that there is almost no human capital accumulation during the stage of physical capital accumulation, the level of capital accumulation in the t-th period of this stage can be calculated by Equation (13) [22]:

$$\overline{K}_t = sP_1 B\overline{K}_{t-1}^{\beta} \tag{13}$$

If steady state is reached in the stage of physical capital, the capital level is set to $\widetilde{K}$:

$$\widetilde{K} = (sP_1 B)^{\frac{1}{1-\beta}} \tag{14}$$

(3) When the economy realizes the first transition, if the steady state level satisfies $\widetilde{K} < K_2$, then before the second transition to the human capital stage, it has reached the steady state of the physical capital development stage. This steady state is more advanced than the previous steady state. In this steady state, both the economic level and the production technology level have reached a higher level than the previous one. If this steady state cannot be overcome and the human capital stage cannot be entered, not only would the economy fail to grow continuously, but similar to the case (1), pollutants $X$ and $Y$ would be produced at the same time. Moreover, it would continue to consume natural and material capital and discharge pollutants and would also face the dilemma of continuous deterioration of environmental quality and stagnation of social operation. This point would continue to be analyzed in the next section, as shown in Figure 3B.

### 2.3.3. Dynamic Analysis of Environmental Quality Changes and the Impact of Economic Transformation Paths

The self-purification of the environment is achieved by relying on natural capital such as trees and water resources. The greater the amount of these natural capitals, the higher

the level of pollutants that can be accommodated and the stronger the self-purification ability. The substantial reason based on environmental problems is because environmental capacity is the connection point between environmental self-purification and environmental problems [23,24]. In environmental science, the equation for calculating absolute capacity and annual capacity in environmental capacity is:

$$WQ = M(Ws - B) \tag{15}$$

$$WA = \frac{A'}{A} \times WQ \tag{16}$$

Among them, *WQ* and *WA* are the environmental capacity, *M* is the environmental space medium, *Ws* is the specified value of the environmental standard, *B* is the environmental background value, and $\frac{A'}{A}$ is the annual self-purification rate.

According to this idea, this paper converts Equation (15) into an "economic approach" and adds it to the dynamic change equation of environmental quality, so that the degradation properties of natural capital can be brought into the framework of transformation analysis. The environmental quality change equation is set as:

$$\dot{E} = -P + \theta W = -\varsigma K + \theta(E_{MAX} - \rho K) \tag{17}$$

Among them, $E_{MAX}$ is the environmental background value, that is, the natural capital stock in the absence of environmental pollution; $\theta$ is the annual self-purification rate, *P* is the pollutant discharge, and $\varsigma$ is the function of capital; *W* is the annual environmental capacity of pollutants. The environmental capacity would be reduced due to the consumption of natural capital or material capital. Therefore, according to the analysis of Equation (17), it is set as a function of natural (material) capital. It is assumed here that the same pollutants are produced per unit of natural capital and physical capital input and have the same impact on environmental capacity. So far, the degradation ability of the natural environment has been introduced into the environmental equation [25].

Assuming that Equation (17) is 0, that is, the number of pollutants that can be self-purified by the environment is equal to the pollutants emitted, the capital stock at this time is:

$$\dot{E} = -P + \theta W = -\varsigma K + \theta(E_{MAX} - \rho K) = 0 \tag{18}$$

Then

$$K_p = \frac{\theta E_{MAX}}{\varsigma + \theta \rho} \tag{19}$$

The dynamic changes of the environment are represented by Figure 4:

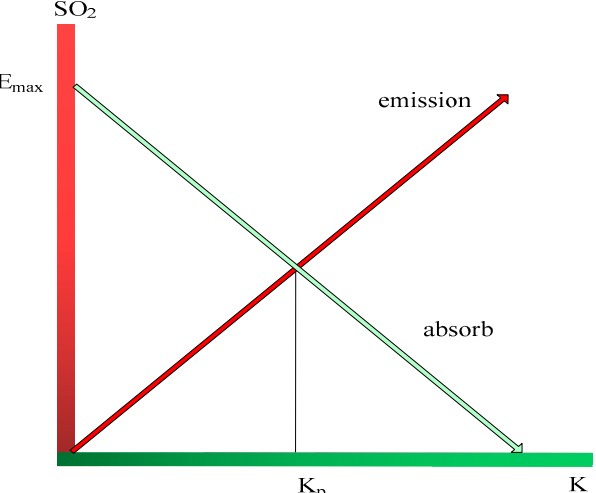

**Figure 4.** Environmental dynamic change diagram.

With economic growth, more and more natural (material) capital is consumed, while the amount of natural capital used for purification in nature is getting smaller and smaller, resulting in a decrease in degradation capacity. When the capital stock is $\overline{K} > K_p$, it is not enough to absorb all the pollutants, and eventually exceeds the load of the environment, causing more and more pollutants to accumulate, causing environmental problems.

## 3. Model Design of the Impact of Environmental Economic Transformation on Financial Eco-Efficiency

### 3.1. Eco-Efficiency Evaluation Method and Its Selection

The measurement of eco-efficiency is the process of concrete measurement of the eco-efficiency of the evaluation object. From the above review of eco-efficiency evaluation methods, it can be found that some scholars use qualitative methods to measure eco-efficiency, some scholars use quantitative methods to measure eco-efficiency, and some scholars use a combination of qualitative and quantitative methods to measure eco-efficiency [26]. The factors considered in the quantitative method are too single, which is only applicable to the analysis of independent and discontinuous research objects and cannot reflect the differences in ecological efficiency of research objects under different environmental conditions. The combination of qualitative and quantitative methods to measure economic benefits can better measure the authenticity of data from a scientific perspective.

In general, the single ratio method refers to the ratio of economic value to environmental impact. When selecting economic value indicators, scholars tend to choose financial indicators with strong availability to reflect the output of enterprises. However, for different research objects, selecting appropriate economic value indicators can not only show the economic value of the research objects, but also highlight the pertinence of the research [27]. However, the factors considered by this method are too single, and it is only suitable for analyzing independent and discontinuous research objects and cannot reflect the differences in the eco-efficiency of the research objects under different environmental conditions, and the calculation results have no grade discrimination, so it cannot reasonably evaluate the pros and cons of eco-efficiency.

In terms of indicators, this paper selects four indicators of environmental economic transformation: return on assets, gross profit rate of sales, period expense rate and total asset turnover rate. Through data analysis, this paper can discuss the impact of these four indicators on financial ecological efficiency, which can be used to predict environmental economic transformation and financial ecological efficiency. This paper selected return on assets (ROA) to measure the financial performance of listed companies in heavily polluting industries [28].

To sum up, this paper first chooses Return on Assets (ROA) as the index to measure the financial performance of listed companies in the heavily polluted industry. Then, by analyzing the impact of ecological efficiency on ROA, GMS, PER and TAT, respectively, this paper discusses the direction and mechanism of the impact of ecological efficiency on financial performance. Financial performance evaluation indicators are shown in Table 1.

**Table 1.** Financial Performance Evaluation Metrics.

| Serial Number | Evaluation Indicators | |
|:---:|:---:|:---:|
| 1 | Return on Assets (ROA) | Financial Performance Indicators |
| 2 | Gross Margin of Sales (GMS) | |
| 3 | Periodic Expense Rate (PER) | Intermediate indicator |
| 4 | Total Asset Turnover (TAT) | |

A reasonable selection of input indicators, expected output indicators, and undesired output indicators is the key to using the undesired output Super-SBM model to measure eco-efficiency. In the eco-efficiency evaluation of listed companies in heavily polluting industries, the connotation of "input" refers to the resources invested and consumed by

the listed company in the production and operation process. "Expected output" refers to the income or profit created by the listed company through the provision of products or services during the production and operation of the listed company. "Unexpected output" refers to the environmental load caused by the listed company's production and operation process. The specific main input and output indicators of the enterprise are shown in Table 2.

**Table 2.** Eco-efficiency evaluation table.

| Index | Specific Indicators | | Model | Serial Number |
|---|---|---|---|---|
| input indicator | resources invested | Total assets | $x_1$ | 1 |
| | | Main business cost | $x_2$ | 2 |
| | Human input | net value of fixed assets | $x_3$ | 3 |
| | | Number of employees | $x_4$ | 4 |
| output indicator | expected output | Main business income | $y_1$ | 5 |
| | | Total profit | $y_2$ | 6 |
| | undesired output | Sulfur dioxide emissions | $y_3$ | 7 |

*3.2. Theoretical Analysis and Research Assumptions*

According to the specific indicators of variable design, the impact of the improvement of eco-efficiency on the return on assets (ROA), gross margin of sales (GMS), period expense rate (PER), and total asset turnover (TAT) are analyzed in turn [29]:

The improvement of eco-efficiency can help enterprises to improve the efficiency of asset use. From the definition of eco-efficiency, it can be seen that enterprise eco-efficiency is a comprehensive evaluation index. The level of this indicator fully reflects all aspects of the comprehensive management of the enterprise, and the return on assets is usually an important indicator used to reflect the effectiveness of the comprehensive management of the enterprise. Therefore, this paper proposes the hypothesis:

Enterprises with higher eco-efficiency can help improve the gross margin of sales of enterprise sales. Enterprises with high eco-efficiency generally have good social reputation, and customers' trust in the company would also be improved, and enterprises can obtain a variety of competitive advantage resources from it. It provides favorable conditions for enterprises to seize the opportunity to open up the market and expand their own scale, so as to win more sales for the company. Therefore, this paper proposes the hypothesis:

Eco-efficient businesses can help reduce financing costs. Enterprises with high eco-efficiency have stronger awareness of environmental protection and social responsibility [30]. First of all, from the perspective of the company, a company with a strong sense of social responsibility is more likely to attract and motivate employees. Under the incentives, employees perform their duties and perform their duties with due diligence, which helps to improve labor productivity, reduce the workload of management personnel, and further make full use of manufacturing costs and management costs. Secondly, from the perspective of investors, companies with a strong sense of social responsibility are also more likely to attract financial investors, thus broadening the company's financing channels and further optimizing the financing environment, which is beneficial for enterprises to save financing costs and reduce financial expenses. Finally, from the perspective of the government, the government departments would appropriately relax the supervision, relax the legal punishment and ease the tax pressure of enterprises with a strong sense of social responsibility, which would also help reduce the company's expenses to a certain extent. Therefore, this paper proposes the hypothesis:

The improvement of eco-efficiency would help enterprises to improve the efficiency of asset management. Specifically, a good social image helps enterprises to establish better cooperative relations, and to a certain extent accelerates the turnover of current assets

such as inventory and receivables. Therefore, this paper proposes the hypothesis. The conceptual framework of hypothetical design variables is shown in Table 3.

**Table 3.** Conceptual framework of hypothetical design variables.

| | |
|---|---|
| $H_1$ | The improvement of eco-efficiency can improve the efficiency of enterprise asset use. |
| $H_{1a}$ | The improvement of eco-efficiency increases the gross margin of sales of the company's sales. |
| $H_{1b}$ | The improvement of eco-efficiency helps to reduce the period cost of enterprises. |
| $H_{1c}$ | The improvement of eco-efficiency helps to accelerate the turnover of enterprise assets. |

## 4. Case Analysis of the Impact of Economic Transformation on Financial Eco-Efficiency

### 4.1. Sample Selection and Data Sources

According to the classification standard of heavy pollution industries, this classification is used as the standard in the sample selection process. First, based on the 16 types of industries included in the heavy polluting industries mentioned above, and referring to the industry classification results of listed companies by China Securities Regulatory Commission in 2017, it screened out listed companies that meet the classification standards for heavy pollution industries; next, the paper intercepted the 2014–2017 panel data of listed companies that met the criteria.

After screening, it was found that the number of listed companies that met the above requirements increased year by year from 2014 to 2020, from 33 in 2014 to 59 in 2020. In order to consider the completeness and comparability of the sample data, this paper finally selects 33 listed companies in heavy pollution industries as the research sample, with a total of 132 observations during the period from 2014 to 2020.

### 4.2. Evaluation Results and Analysis of Eco-Efficiency

Using MaxDEA software, the eco-efficiency evaluation of the 33 listed companies selected in this paper was conducted, and the eco-efficiency evaluation results of the sample companies in each year from 2014 to 2017 were finally obtained, as shown in Figure 5 and Table 4 [31,32]. The data has referred several references.

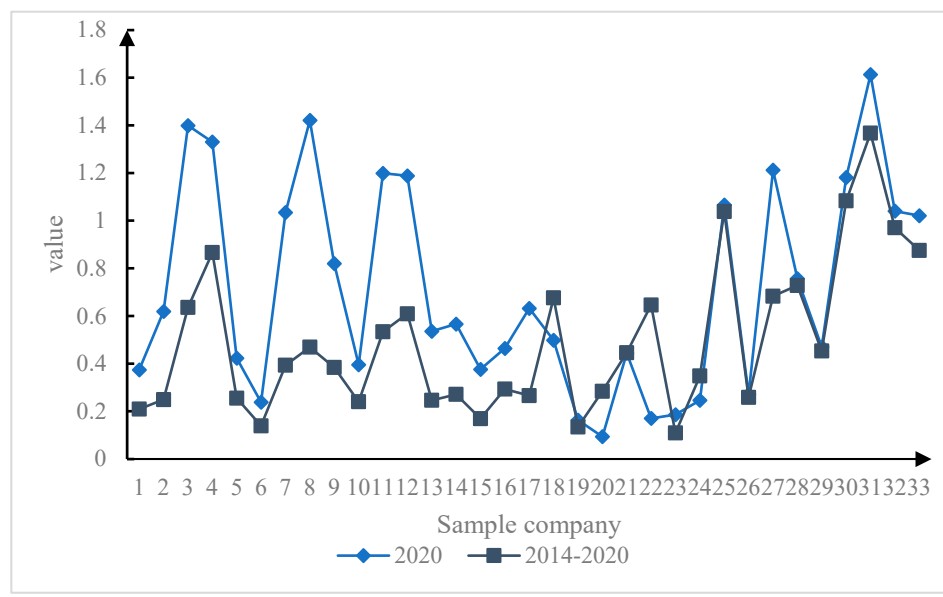

**Figure 5.** Average corporate eco-efficiency values between 2020 and 7 years.

**Table 4.** Eco-efficiency values of 33 companies in 2014–2020.

| Year | Average | Maximum | Minimum |
|------|---------|---------|---------|
| 2014 | 0.468 | 0.968 | 0.100 |
| 2015 | 0.387 | 0.998 | 0.121 |
| 2016 | 0.397 | 1.032 | 0.100 |
| 2017 | 0.705 | 1.612 | 0.345 |
| 2018 | 0.735 | 1.721 | 0.521 |
| 2019 | 0.731 | 1.638 | 0.483 |
| 2020 | 0.712 | 1.852 | 0.557 |

Figure 5 reflects the eco-efficiency values of the 33 companies in 2020 and the average eco-efficiency values of the 33 companies in 7 years. By comparing the average eco-efficiency of 33 companies between 2020 and 7 years, it is obvious that almost all companies' eco-efficiency values are higher than the average. Table 3 reflects the annual changes in the eco-efficiency value of 33 companies over the past 7 years. It can be found that 2017 was a step, and the eco-efficiency value increased significantly. This shows that in recent years, the implementation effect of the energy-saving and emission-reduction policies formulated by the Chinese government has shown. With the strengthening of environmental supervision by the state and the emphasis on environmental governance by enterprises, the overall eco-efficiency level of listed companies in heavily polluting industries is also rising.

*4.3. Descriptive Statistical Analysis*

According to the specific values of return on assets (ROA), gross margin of sales (GMS), period expense rate (PER), total asset turnover (TAT) of listed companies in heavy pollution industries from 2014 to 2020 in the appendix, and the size of the enterprise (SOTE), asset-liability ratio (ALR), equity concentration (EC), and listing years (LY), and perform descriptive statistics to obtain the descriptive statistics of related variables in Figure 6.

Figure 6A shows the minimum value, median and maximum value of each index in 7 years, and Figure 6B shows the mean value and standard deviation of each index in 7 years. The approximate distribution of values can be known by comparing the median to the mean. The following are the specific analysis results:

(1) The average return on assets (ROA) and gross margin of sales (GMS) of each company are 4.580% and 18.354%, respectively, indicating that the average return on sample companies is positive. (2) The median of the period expense rate (PER) of the sample enterprises is less than the mean (9.43011.903). That is, half of the enterprises' period expense rates are lower than the mean, indicating that the sample enterprises' period expense management and control efficiency is high. (3) The average value of the total asset turnover (TAT) of the sample companies is greater than the median (0.7340.519). That is, the total asset turnover of half of the sample companies is less than the average value, indicating that the turnover speed of more than half of the sample companies is relatively slow and the operating capacity is weak. (4) The median of the eco-efficiency (ECO) of the sample enterprises is less than the mean value of eco-efficiency (0.3820.489), indicating that the eco-efficiency of most enterprises does not reach the average level. In addition, there is a big difference between the minimum value of 0.012 and the maximum value of 1.712 for the eco-efficiency of the sample enterprises, indicating that the eco-efficiency levels of the sample enterprises are uneven. (5) The minimum and maximum logarithm of total assets (SOTE) are 9.425 and 11.757, respectively. The mean and standard deviation are 10.693 and 0.539, respectively, indicating that there is little difference in the scale of sample enterprises. The median (67.3%) of the asset-liability ratio (ALR) of the sample enterprises is greater than the mean value of the asset-liability ratio (62.875%), indicating that at least half of the sample enterprises have an asset-liability ratio greater than 62.875%. The average shareholding ratio of the top ten shareholders is 69.254% [33].

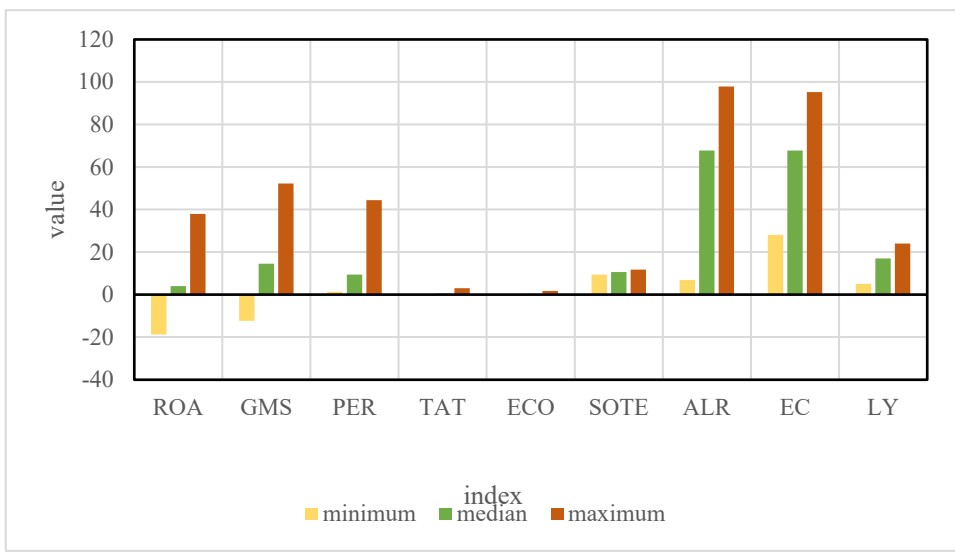

(**A**) Distribution of values

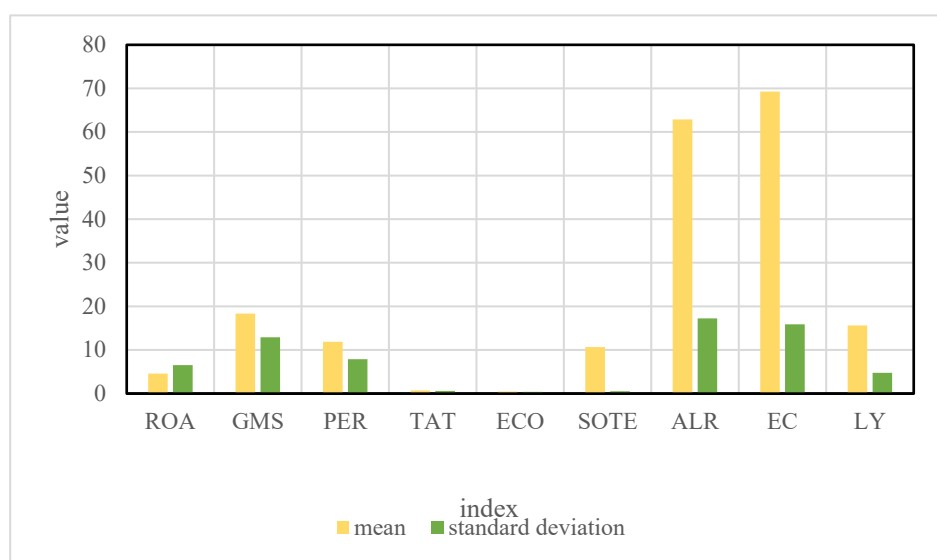

(**B**) The degree of dispersion of values

**Figure 6.** Descriptive statistics of variables.

*4.4. Correlation Analysis*

It can be seen from Table 5 that the Pearson correlation coefficient of return on assets (ROA) and eco-efficiency (ECO) is 0.374, the significance level is 0.000, and there is a significant positive correlation at the 0.1% level. The Pearson correlation coefficient of gross margin of sales (GMS) and eco-efficiency (ECO) is 0.344, the significance level is 0.000, and there is a significant positive correlation at the 0.1% level. The Pearson correlation coefficient between period expense rate (PER) and eco-efficiency (ECO) is −0.007, the significance level is 0.940, and the correlation is weak. The Pearson correlation coefficient of total asset turnover (TAT) and eco-efficiency (ECO) was 0.270, the significance level was 0.002, and there was a significant positive correlation at the 1% level. The above results show that under the Pearson correlation test, the ecological efficiency is significantly positively correlated with the return on assets, gross margin of sales, and total asset turnover, and has a weak negative correlation with the period expense rate.

**Table 5.** Pearson correlation test results.

|  | **ROA** | **GMS** | **PER** | **TAT** |
|---|---|---|---|---|
| Correlation | 0.352 | 0.324 | −0.007 | 0.271 |
| salience | 0.001 | 0.00 | 0.942 | 0.002 |
| N | 135 | 135 | 135 | 135 |

The Pearson correlation coefficient reflects the correlation between any two variables. The addition of a control variable may affect the relationship between the dependent variable and the independent variable. Therefore, the company size, asset-liability ratio, equity concentration and listing years are added as control variables to study the correlation between ecological efficiency and financial performance.

Comparing Table 4 with Table 6, it can be found that return on assets (ROA), gross margin of sales (GMS), and total asset turnover (TAT) have all passed the significance test. It can be seen from Table 6 that ECO and ROA are significantly positively correlated ($p < 0.01$), which preliminarily shows that the improvement of ecological efficiency has a promoting effect on the financial performance indicator ROA. There was a significant positive correlation between ECO and GMS ($p < 0.01$), a significant positive correlation between ECO and TAT ($p < 0.01$), and a negative correlation between ECO and PER, but it was not significant. It shows that the improvement of ecological efficiency can promote the improvement of enterprise financial performance index ROA by affecting the gross margin of sales of sales and total asset turnover rate. It can also promote the improvement of enterprise financial performance index ROA by affecting the period expense rate, but the effect is not obvious.

**Table 6.** Partial correlation test results.

|  | **ROA** | **GMS** | **PER** | **TAT** |
|---|---|---|---|---|
| Correlation | 0.332 | 0.276 | −0.138 | 0.293 |
| salience | 0.000 | 0.0012 | 0.123 | 0.001 |
| DF | 126 | 126 | 126 | 126 |

*4.5. Panel Data Regression Analysis*

According to the article, four panel data models of the impact of environmental economic transformation on ecological efficiency were constructed to test the direction and extent of the impact of environmental economic transformation on ecological efficiency. It was originally assumed $all \alpha_i = 0$, that is, the intercept term of all individuals is the same. When the obtained $p$ value is 0.0000, the null hypothesis is strongly rejected. The fixed effect model is selected, otherwise, the random effect model is selected. From the Hausman test results in Table 7, it can be found that the $p$-value obtained by model (3-3) and model (3-5) is 0.0000, rejecting the null hypothesis and choosing the fixed effect model, while model (3-2) and model (3-4) choose a random effects model.

**Table 7.** Hausman test results.

| **Model** | **Index** | **Hausman Test** |
|---|---|---|
| (3-2) | ROA | 0.608 |
| (3-3) | GMS | 0.000 |
| (3-4) | PER | 0.128 |
| (3-5) | TAT | 0.000 |

In order to study the direction and extent of the impact of ecological efficiency on financial performance of listed companies in heavily polluting industries, according to the

model constructed in this paper, Stata12.0 software is used to perform regression analysis on relevant variables and data. The regression results of the four panel data models are as follows:

(1) Influence of return on assets and gross margin of sales on ecological efficiency

In this paper, Stata12.0. software is used to perform regression analysis on panel data model 3-2 and model 3-3 to explore the impact of ecological efficiency on asset returns. The analysis results are shown in Figure 7.

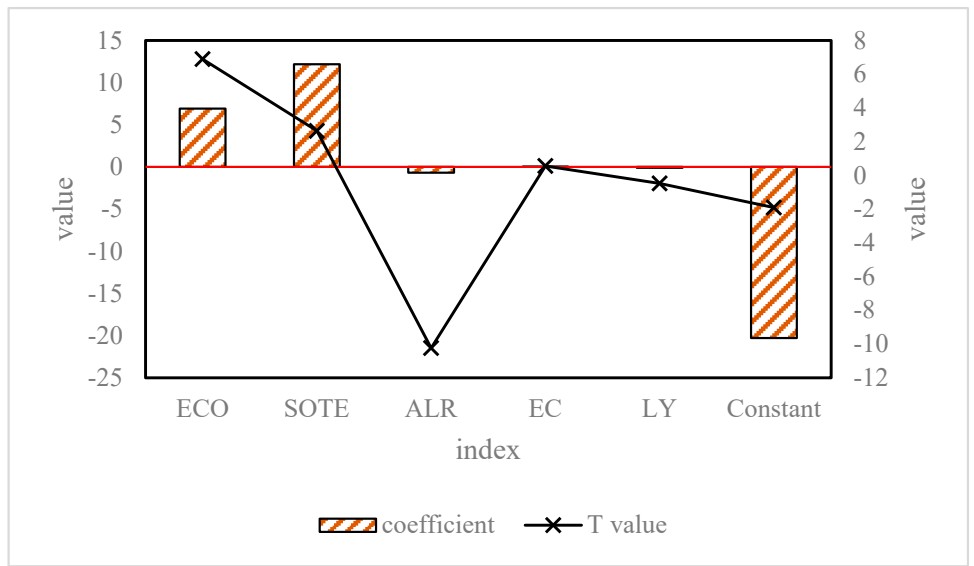

(**A**) Model 3-2

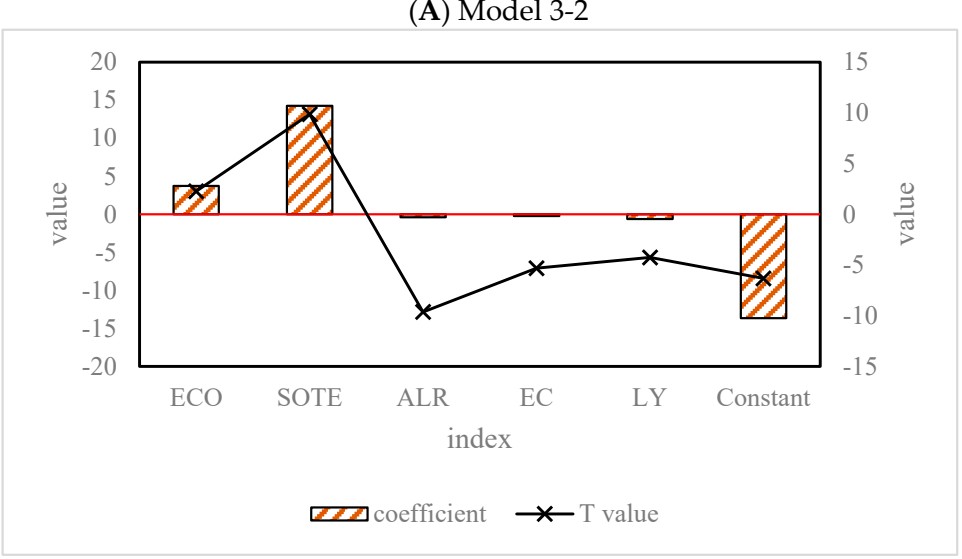

(**B**) Model 3-3

**Figure 7.** The impact of return on assets and gross sales margin on eco-efficiency.

It can be seen from Figure 7A that ECO and return on assets (ROA) are significantly positively correlated ($p < 0.01$), indicating that the improvement of ecological efficiency helps to improve the efficiency of asset use, that is, the improvement of ecological efficiency can help to improve the investment efficiency of listed companies, and the hypothesis $H_1$ is verified. At the same time, it can also be found that the regression coefficient of enterprise size is 12.162 ($p < 0.01$), indicating that enterprise size as a control variable has a

significant positive correlation with the return on assets. The regression coefficient of the asset-liability ratio is $-0.683$ ($p < 0.01$), indicating that the asset-liability ratio as a control variable is significantly negatively correlated with the return on assets. The concentration of ownership, listing years and the return on assets are not correlated.

It can be seen from Figure 7B that ECO is significantly positively correlated with gross margin of sales (GMS) ($p < 0.05$), indicating that the higher the eco-efficiency, the greater the company's gross margin of sales, which verifies Hypothesis 1. At the same time, it can also be found that the regression coefficient of enterprise scale is 14.5858 ($p < 0.01$), indicating that enterprise scale as a control variable has a significant positive correlation with gross margin of sales. The regression coefficients of the asset-liability ratio, equity concentration, and listing years are $-0.413$ ($p < 0.01$), $-0.285$ ($p < 0.01$), and $-0.637$ ($p < 0.01$), respectively. This indicates that the asset-liability ratio, equity concentration and listing years as control variables are significantly negatively correlated with gross margin of sales.

(2) Periodic expense rate and its influence on total asset turnover rate on ecological efficiency

In this paper, Stata12.0. software is used to conduct regression analysis on panel data model 3-4 and model 3-5 to explore the impact of period expense rate and total asset turnover rate on ecological efficiency. The analysis results are shown in Figure 8.

It can be seen from Figure 8A that ECO is significantly negatively correlated with the period expense ratio (PER) ($p < 0.05$), indicating that the improvement of eco-efficiency can help reduce the current period expense of the enterprise, which verifies Hypothesis $H_{1b}$. At the same time, it can also be found that the regression coefficients of enterprise scale and asset-liability ratio are 10.7839 ($p < 0.10$) and 0.1364 ($p < 0.01$), respectively. This indicates that the scale of enterprises and the asset-liability ratio as control variables are significantly positively correlated with the period expense rate. The equity concentration, listing years, and the period expense rate are not correlated.

It can be seen from Figure 8B that ECO is significantly positively correlated with the total asset turnover (TAT) ($p < 0.01$), indicating that the improvement of ecological efficiency helps the turnover of assets. The improvement of ecological efficiency can help improve the utilization efficiency of listed companies' assets, which verifies the hypothesis $H_{1c}$. At the same time, it can also be found that the regression coefficient of enterprise size is $-0.4650$ ($p < 0.01$), indicating that enterprise size as a control variable is significantly negatively correlated with total asset turnover. The asset turnover rate is not related to the total asset turnover. The regression coefficients of equity concentration and listing years are 0.0084 ($p < 0.01$) and 0.0115 ($p < 0.05$), respectively, indicating that the equity concentration and listing years as control variables are significantly positively correlated with the total asset turnover.

According to the four panel data models of the impact of environmental and economic transformation on ecological efficiency constructed in the article, and combining the four indicators of ROA, GMS, PER, and TAT, the comprehensive coefficient of the environmental, and economic transformation indicators calculated by the software is 1.325 ($p < 0.001$). The experimental results are helpful for listed companies in the heavily polluted industry to actively explore ways to improve ecological efficiency.

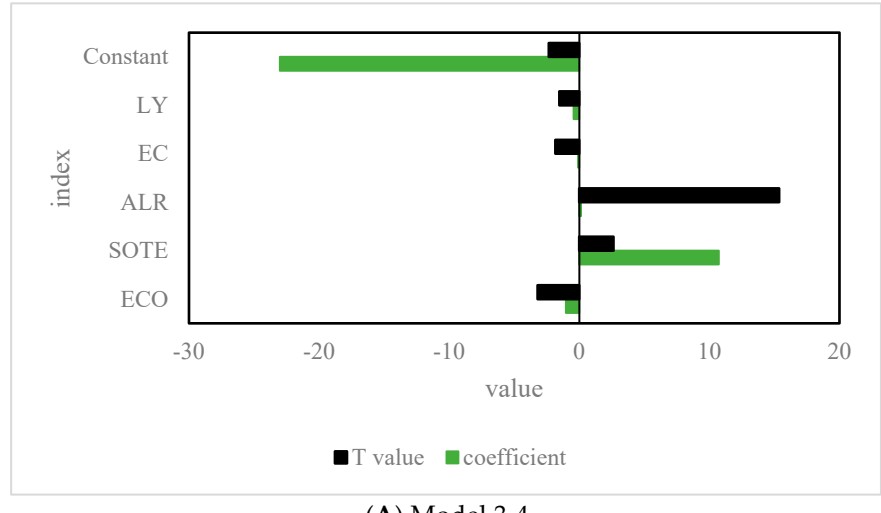

(**A**) Model 3-4

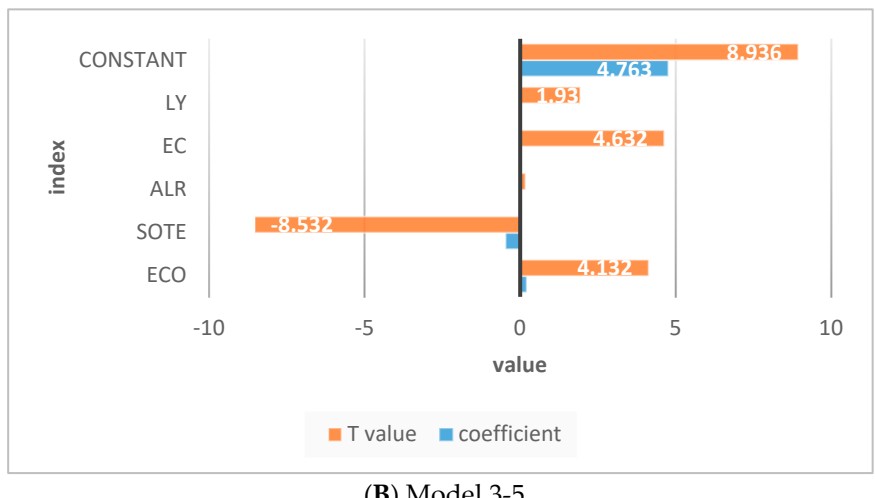

(**B**) Model 3-5

**Figure 8.** Periodic Expense Rate and Impact on Total Asset Turnover on Eco-efficiency.

## 5. Discussion

This paper aimed to explore the path of environmental economic transformation and analyze the impact of environmental economic transformation on financial ecological efficiency. Firstly, it introduced the transformation of environmental economy, including the evaluation of ecological efficiency, the green transformation of economy, environmental quality, and the growth path of economic transformation. Then it introduced the model design of the impact of environmental economic transformation on financial ecological efficiency, including ecological efficiency evaluation methods and their choices, theoretical analysis, and research assumptions. Finally, this paper introduced the case study of the impact of economic transformation on financial ecological efficiency. In terms of indicators, this paper selected four indicators of environmental economic transformation: return on assets, gross profit rate of sales, period expense rate, and total asset turnover rate. This paper discussed the impact of these four indicators on financial ecological efficiency in depth. Of course, this paper also has shortcomings. It did not compare the overall analysis logic and path of economic transformation and environmental quality constructed in this paper with the traditional system, which led to a conclusion that may not be very scientific. In future work, researchers should pay attention to conducting comprehensive comparative tests.

## 6. Conclusions

In order to improve the ecological efficiency of listed companies in heavily polluting industries, in addition to strengthening guidance and constraints from the government, companies themselves must also attach great importance to it. Based on the ecological efficiency evaluation system and evaluation model, this paper evaluated and analyzed the ecological efficiency of listed companies in heavily polluting industries. Through empirical research, this paper studied the impact and impact mechanism of ecological efficiency on financial performance and proved that the improvement of ecological efficiency has a promoting effect on the improvement of corporate financial performance. This conclusion is helpful for listed companies in heavily polluting industries to actively explore ways to improve ecological efficiency. Therefore, according to the input-output indicators in the eco-efficiency evaluation system, this paper proposed countermeasures and suggestions to improve the eco-efficiency of heavily polluting industries from four perspectives: resource input, human input, expected output, and undesired output. Among them, "resource utilization" is the source for enterprises in heavily polluting industries to improve ecological efficiency; "quality of talents" is the driving force for enterprises in heavy pollution industries to improve ecological efficiency; "economic growth" is the goal of enterprises in heavy pollution industries to improve ecological efficiency; and "reducing pollution" is the basis for enterprises in heavily polluting industries to improve their ecological efficiency. Enterprises in heavy pollution industries can realize the sustainable development of enterprises in heavy pollution industries by strengthening and improving these four aspects.

**Author Contributions:** L.Y. and J.L. designed and performed the experiment and prepared this manuscript. All coauthors contributed to manuscript editing. All authors have read and agreed to the published version of the manuscript.

**Funding:** This research was funded by Heilongjiang Province Philosophy and Social Science Research Planning Project. Project Name: Research on Green Finance Development in Heilongjiang Province. Project Number: 21JYB142.

**Institutional Review Board Statement:** Not applicable.

**Informed Consent Statement:** Not applicable.

**Data Availability Statement:** The data that support the findings of this study are available from the corresponding author upon reasonable request.

**Conflicts of Interest:** The authors declare no conflict of interest.

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
