# Peer review of "Impact of Environmental Economic Transformation Based on Sustainable Development on Financial Eco-Efficiency"

_sustainability, doi:10.3390/su15010856_

Round 1
Reviewer 1 Report
The paper's topic and conducted research are very important and justified to be presented in a high-quality Journal. The subject is very important for the literature. The paper is informative, but some issues need to be addressed carefully. My decision is a major revision with some amendments. Please see my comments and suggestions below.
Comment 1. In the introduction, the authors should have described in more detail on research gap.
Comment 2. Also, in the introduction, I think that the authors should include more references about the relationship between sustainable development and finance. These are some useful papers:
-Qing, L.; Chun, D.; Dagestani, A.A.; Li, P. Does Proactive Green Technology Innovation Improve Financial Performance? Evidence from Listed Companies with Semiconductor Concepts Stock in China. Sustainability 2022,14, 4600. doi:10.3390/su14084600
-Qing, L., Alwahed Dagestani, A., Shinwari, R., & Chun, D. (2022). Novel research methods to evaluate renewable energy and energy-related greenhouse gases: evidence from BRICS economies. Economic Research-Ekonomska Istraživanja, 1-17. doi: https://doi.org/10.1080/1331677X.2022.2080746
Comment 3. Figure 2 is not clearly expressed. Please revise it.
Comment 4. The methodological approach is too vague. You must extend your paper with a brief statement about your methodological approach and empirical model.
Comment 5. In section 3.1, Financial Performance Evaluation Metrics and Eco-efficiency evaluation should be supported by previous literature. Please explain in detail why these indicators need to be selected.
Comment 6. In section 3.2, the hypothesis should be supported by more previous literature, including H1, H1a, H1b, and H1c. In addition, the authors should add a conceptual framework for the study.
Comment 7. And discussion? The authors should add the section discussion. Discussion should be the evaluation of the obtained results (author's original thoughts) in the light of the previous research (could include the items as the explanation of the hypothesis). Also, the authors should add theoretical contributions, managerial implications, limitations, and future research.
Comment 8. The number of references is not adequate.
Good luck for your work!
Author Response
The paper's topic and conducted research are very important and justified to be presented in a high-quality Journal. The subject is very important for the literature. The paper is informative, but some issues need to be addressed carefully. My decision is a major revision with some amendments. Please see my comments and suggestions below.
Comment 1. In the introduction, the authors should have described in more detail on research gap.
Answer: The ecological efficiency of various regions in China shows a positive spatial correlation feature, and there is an obvious spatial aggregation feature.
Comment 2. Also, in the introduction, I think that the authors should include more references about the relationship between sustainable development and finance. These are some useful papers:
-Qing, L.; Chun, D.; Dagestani, A.A.; Li, P. Does Proactive Green Technology Innovation Improve Financial Performance? Evidence from Listed Companies with Semiconductor Concepts Stock in China. Sustainability 2022,14, 4600. doi:10.3390/su14084600
-Qing, L., Alwahed Dagestani, A., Shinwari, R., & Chun, D. (2022). Novel research methods to evaluate renewable energy and energy-related greenhouse gases: evidence from BRICS economies. Economic Research-Ekonomska Istraživanja, 1-17. doi: https://doi.org/10.1080/1331677X.2022.2080746
Answer: thank your very much for your comments, i have added these references to the paper,
Comment 3. Figure 2 is not clearly expressed. Please revise it.
Answer: thank you for your comment, i have deleted the Figure 2 and depict the content in words in this paper.
Comment 4. The methodological approach is too vague. You must extend your paper with a brief statement about your methodological approach and empirical model.
Answer: i have revised the conclusion in the method of research as follows: To sum up, this paper first chooses Return on Assets (ROA) as the index to measure the financial performance of listed companies in the heavily polluted industry. Then, by analyzing the impact of ecological efficiency on ROA, GMS, PER and TAT respectively, this paper discusses the direction and mechanism of the impact of ecological efficiency on financial performance. Financial performance evaluation indicators are shown in Table 1.
Comment 5. In section 3.1, Financial Performance Evaluation Metrics and Eco-efficiency evaluation should be supported by previous literature. Please explain in detail why these indicators need to be selected.
Answer: reason to selecting these indicators: In terms of indicators, this paper selects four indicators of environmental economic transformation: return on assets, gross profit rate of sales, period expense rate and total asset turnover rate. Through data analysis, this paper can discuss the impact of these four indicators on financial ecological efficiency, which can be used to predict environmental economic transformation and financial ecological efficiency.
Reference added: [28] Devi, H. P. . "Pengaruh Rasio Kesehatan Bank (CAR, NPF, FDR, BOPO) Terhadap Return On Assets pada Bank Umum Syariah di Indonesia." Owner 5.1(2021):1-11.
Comment 6. In section 3.2, the hypothesis should be supported by more previous literature, including H1, H1a, H1b, and H1c. In addition, the authors should add a conceptual framework for the study.
Answer: thank you for you comment, i have added it in Table :Conceptual framework of hypothetical design variables
Comment 7. And discussion? The authors should add the section discussion. Discussion should be the evaluation of the obtained results (author's original thoughts) in the light of the previous research (could include the items as the explanation of the hypothesis). Also, the authors should add theoretical contributions, managerial implications, limitations, and future research.
Answer: thank you very much for your suggestion, i have added the part of Discussion in this paper. “This paper aimed to explore the path of environmental economic transformation and analyze the impact of environmental economic transformation on financial ecological efficiency. Firstly, it introduced the transformation of environmental economy, including the evaluation of ecological efficiency, the green transformation of economy, environmental quality and the growth path of economic transformation. Then it introduced the model design of the impact of environmental economic transformation on financial ecological efficiency, including ecological efficiency evaluation methods and their choices, theoretical analysis and research assumptions. Finally, this paper introduced the case study of the impact of economic transformation on financial ecological efficiency. In terms of indicators, this paper selected four indicators of environmental economic transformation: return on assets, gross profit rate of sales, period expense rate and total asset turnover rate. This paper discussed the impact of these four indicators on financial ecological efficiency in depth. Of course, this paper also has shortcomings. It did not compare the overall analysis logic and path of economic transformation and environmental quality constructed in this paper with the traditional system, which leaded to the conclusion that may not be very scientific. In future work, it should pay attention to carrying out comprehensive comparative tests.”
Comment 8. The number of references is not adequate.
Good luck for your work!
Answer: thank you for your comment, i have added several references in the paper, please check it.
Reviewer 2 Report
Please see the attachment.

Author Response
The authors analyze an important problem, which plays a role in promoting the
development of sustainable development field. However, there are still some problems in
the paper. The authors are recommended to carry out a major revision according to the
following suggestions:
Paper Structure:
- Please revise the introduction according to the standard format: research background,
research motivation, literature review, this paper's contribution and the following
structure. It is beneficial for readers if you can use a chart of the paper structure to
show it clearly.
Answer:Thank your for your comment, i have rewritten the part of Introduction according to your request as follows: It has been nearly 40 years since the reform and opening up, and China's economy is also in the transition stage from high-speed to medium high-speed and high-quality. In this context, the current environmental economic transformation lacks the corresponding evaluation model, so this paper aims to explore the path of environmental economic transformation and analyze the impact of environmental economic transformation on financial ecological efficiency. From the perspective of economic stage transformation, this paper combs the existing theoretical research and literature achievements with the main line of "transformation growth and environmental financial ecological efficiency green consumption transformation". It probes into the relationship between natural environment and economic development in the process of transformation.
- The length of each Section should be moderate. In this paper, section 2 and 4 is too
long, while Introduction is relatively short. The author should delete or integrate the
contents of the sections.
Answer: I have adjusted the length of each paragraph to make the layout of this paper more reasonable.
- There are a few grammar, phrases and typos in the paper, please check carefully.
Answer: glad to receive your comments, i have revised and corrected these problems, please check it.
- Please ensure that the author has the right to use some pictures, such as Figure 2. It
looks like they are downloaded from the Internet.
Answer: glad to receive your suggest, i have deleted Figure 2, please check it.
- The format of the figure should be consistent, such as the two figures in Figure 9.
Answer: thank you for your comments,i have keep the format in consistent, please check it.
Empirical Design:
- At present, the interpretation of data and variables in this paper is not enough. The
author needs to strengthen the description of the source of indicators and the method
of data measurement.
Answer: glad to receive your comments,i have added related method of data measurement as follows: This paper used Correlation Analysis. Diminishing marginal utility refers to that when a person continuously consumes a certain kind of goods within a certain period of time, while the consumption quantity of other goods remains unchanged, the total utility will increase with the increase of the quantity of the goods consumed. However, marginal utility of the goods (that is, increase of the utility brought by each unit of the goods consumed) tends to decrease.
- In the absence of a detailed description of empirical methods, it is necessary to provide
more references and explanations. Suggested references in Efficiency measurement
(DEA):
(1) Spatial Interaction Spillover Effects between Digital Financial Technology and Urban
Ecological Efficiency in China: An Empirical Study Based on Spatial Simultaneous
Equations. International Journal of Environmental Research and Public Health 18, 8535.
(2) Green Total Factor Productivity Growth: Policy-Guided or Market-Driven?
International Journal of Environmental Research and Public Health 19, 10471.
(3) How do environmental regulation and environmental decentralization affect green
total factor energy efficiency: Evidence from China. Energy Economics, 91, 104880.
Answer: thank you very much your your suggestion, i have added them in the absence of a detailed description of empirical methods to make the paper more logical.
- In the method part of this paper, there are too many descriptions of some common
formulas, which should be appropriately deleted and supplemented with relevant
references.
Answer: thanks a lot for your suggestion, i have revised them according to your request.
- The part of introducing the method should strengthen the connection with the research
object of this paper, so as not to be incompatible with the overall structure of this paper.
Answer:The factors considered in quantitative method are too single, which is only applicable to the analysis of independent and discontinuous research objects, and cannot reflect the differences in ecological efficiency of research objects under different environmental conditions. The combination of qualitative and quantitative methods to measure economic benefits can better measure the authenticity of data from a scientific perspective.
- Does the result value have any economic implications? The author needs to make a
supplementary explanation.
Answer: The experimental results are helpful for listed companies in the heavily polluted industry to actively explore ways to improve ecological efficiency.
Theoretical Analysis:
- It is necessary to add discussion at the end of the paper, which can be placed behind
the analysis or in the same section as the conclusion. The contents of the discussion
include: in-depth analysis of the conclusions of this paper, the shortcomings of this
paper and the prospects for future research. A good "discussion" can make readers
deeply understand the content of this paper and improve the contribution of this paper.
Answer: thank you very much for your suggestion,i have added part of discussion as follows: This paper aims to explore the path of environmental economic transformation and analyze the impact of environmental economic transformation on financial ecological efficiency. Firstly, it introduces the transformation of environmental economy, including the evaluation of ecological efficiency, the green transformation of economy, environmental quality and the growth path of economic transformation; Then it introduces the model design of the impact of environmental economic transformation on financial ecological efficiency, including ecological efficiency evaluation methods and their choices, theoretical analysis and research assumptions. Finally, this paper introduces the case study of the impact of economic transformation on financial ecological efficiency. In terms of indicators, this paper selects four indicators of environmental economic transformation: return on assets, gross profit rate of sales, period expense rate and total asset turnover rate. This paper discusses the impact of these four indicators on financial ecological efficiency in depth. Of course, this paper also has shortcomings. It does not compare the overall analysis logic and path of economic transformation and environmental quality constructed in this paper with the traditional system, which leads to the conclusion that may not be very scientific. In future work, we should pay attention to carrying out comprehensive comparative tests.
Round 2
Reviewer 1 Report
I appreciate the authors' effort to improve the paper. After a carefully reviewing your revised manuscript, I am highly satisfied with the changes that you have made. However, I still suggest that the authors could enrich and extend the managerial implications. Also, widening the limitations of this research, because they are not only those mentioned by the authors. I can recommend the publication of this research. I wish you well in taking your research forward.

Reviewer 2 Report
Accept in present form